# Cardio-centric hemodynamic management improves spinal cord oxygenation and mitigates hemorrhage in acute spinal cord injury

Alexandra M. Williams [1,2], Neda Manouchehri[1], Erin Erskine[1,2], Keerit Tauh[1], Kitty So[1], Katelyn Shortt[1], Megan Webster[1], Shera Fisk[1], Avril Billingsley[1], Alex Munro[1], Seth Tigchelaar [1], Femke Streijger[1], Kyoung-Tae Kim[1,3], Brian K. Kwon[1,4,5] & Christopher R. West [1,2,5✉]

Chronic high-thoracic and cervical spinal cord injury (SCI) results in a complex phenotype of cardiovascular consequences, including impaired left ventricular (LV) contractility. Here, we aim to determine whether such dysfunction manifests immediately post-injury, and if so, whether correcting impaired contractility can improve spinal cord oxygenation ($SCO_2$), blood flow (SCBF) and metabolism. Using a porcine model of T2 SCI, we assess LV end-systolic elastance (contractility) via invasive pressure-volume catheterization, monitor intraparenchymal $SCO_2$ and SCBF with fiberoptic oxygen sensors and laser-Doppler flowmetry, respectively, and quantify spinal cord metabolites with microdialysis. We demonstrate that high-thoracic SCI acutely impairs cardiac contractility and substantially reduces $SCO_2$ and SCBF within the first hours post-injury. Utilizing the same model, we next show that augmenting LV contractility with the β-agonist dobutamine increases $SCO_2$ and SCBF more effectively than vasopressor therapy, whilst also mitigating increased anaerobic metabolism and hemorrhage in the injured cord. Finally, in pigs with T2 SCI survived for 12 weeks post-injury, we confirm that acute hemodynamic management with dobutamine appears to preserve cardiac function and improve hemodynamic outcomes in the chronic setting. Our data support that cardio-centric hemodynamic management represents an advantageous alternative to the current clinical standard of vasopressor therapy for acute traumatic SCI.

---

[1] International Collaboration On Repair Discoveries (ICORD), University of British Columbia, Vancouver, BC, Canada. [2] Department of Cellular and Physiological Sciences, Faculty of Medicine, University of British Columbia, Vancouver, BC, Canada. [3] Department of Neurosurgery, School of Medicine, Kyungpook National University Hospital, Daegu, South Korea. [4] Vancouver Spine Surgery Institute, Department of Orthopaedics, University of British Columbia, Vancouver, BC, Canada. [5]These authors contributed equally: Brian K. Kwon, Christopher R. West. ✉email: chris.west@ubc.ca

 1

The acute phase following a traumatic spinal cord injury (SCI) represents an important therapeutic window of opportunity to intervene with neuroprotective approaches that might limit secondary damage to the injured cord[1], thereby providing the patient with the best chance of attaining some neurological recovery. Hemodynamic management is one of the only neuroprotective strategies available to clinicians, and current guidelines suggest that mean arterial pressure (MAP) be maintained between 85 and 90 mmHg with intravenous fluids and vasopressors such as norepinephrine (NE), with the aim of offsetting systemic hypotension and maintaining adequate spinal cord perfusion[2]. Though this "one-size-fits-all" strategy can improve spinal cord blood flow (SCBF), vasopressor management with NE has been shown to produce potentially harmful SCBF profiles in some acute SCI patients[3] and has been shown by multiple investigators to exacerbate intraparenchymal hemorrhage[4–6]. In the setting of acute SCI, clinical studies have shown strong associations between increased cord hemorrhaging and worsened neurological outcomes (i.e. more neurologically complete injuries)[7]. Such hemorrhaging is therefore a critically concerning side-effect in the application of vasopressor therapy as a first-line hemodynamic management strategy for patients with acute SCI.

To date, the clinical literature has not considered that cardiac contractile dysfunction may occur acutely post-SCI and contribute to reduced spinal cord oxygenation ($SCO_2$) and SCBF. As such, a hemodynamic management strategy that focuses on the heart has not been forthcoming. Only a single published study has considered the use of inotropic agents such as dobutamine (DOB) for hemodynamic management of acute SCI[8], however the efficacy of cardiac inotropes in improving cardiovascular and spinal cord hemodynamics has not been directly compared with that of vasopressor-based management strategies that focus solely on MAP.

Accordingly, the aims of the current research were threefold. In experiment 1, we sought to define the acute impact of high-level SCI on left ventricular (LV) contractility (i.e. end-systolic elastance; $E_{es}$) using our porcine model of contusion SCI at the second thoracic spinal cord level (T2)[9]. In experiment 2, we conducted a randomized intervention trial in the same porcine model to compare the efficacy of using the cardiac β-agonist DOB versus NE (i.e., current clinical standard) in augmenting $SCO_2$ and SCBF acutely following T2 SCI. In experiment 3, we assessed whether acute management with DOB or NE generates favourable chronic cardiac and hemodynamic outcomes in animals with T2 SCI. To address the aims of experiments 1 and 2, a total of 22 female Yucatan minipigs were instrumented with a LV pressure-volume admittance catheter and Swan-Ganz catheter (Fig. 1a), as well as intraparenchymal probes for $SCO_2$, SCBF, and microdialysis placed 1.2 cm and 3.2 cm caudal to the site of injury. Animals received a T2 contusion injury (50 g weight drop, ~16 cm height) with 2 h compression (150 g total), and hemodynamic management with DOB or NE began 30 mins post-SCI up until 4 h post-SCI (experimental endpoint). In experiment 3, we conducted a proof-of-principle study to assess the long-term changes in cardiovascular hemodynamics associated with acute DOB administration. Ten Yucatan minipigs with acute T2 SCI were randomized to receive acute hemodynamic management with DOB or NE for 6 h post-SCI, and were subsequently survived for 12 weeks. Animals were instrumented with a Swan-Ganz catheter during both injury and outcome surgeries (i.e. 12 weeks post-SCI), as well as a LV pressure-volume catheter during outcome surgery. Here, we demonstrate first that LV load-independent contractile function, including $E_{es}$, is impaired within the first hour following a T2 SCI. We thereafter find that treating the reduced contractility with DOB is more efficacious

than NE with respect to optimizing hemodynamics, improving the spinal cord microenvironment, and reducing intraparenchymal hemorrhage. Importantly, we further establish the potential long-term benefits of cardio-centric hemodynamic management by experimentally demonstrating that DOB treatment, but not NE, preserves cardiac function and normalizes blood pressure in a chronic model of T2 SCI.

## Results

**Cardiac contractility is impaired in acute T2 SCI.** In experiment 1 ($n = 8$), LV maximal systolic pressure ($P_{max}$), MAP and total peripheral resistance (TPR) were reduced within 1 h post-SCI, and remained depressed up to 4 h post-SCI (Fig. 1b–d, i). At 4 h post-injury there was a slight but significant increase to LV filling volume (i.e. EDV; Fig. 1e); however, there were no significant alterations to LV stroke volume (SV, Fig. 1f), ejection fraction (EF, Fig. 1g), cardiac output measured with thermodilution ($Q_{TD}$, Fig. 1h), or heart rate (Supplementary Table 1) within the 4 h following T2 SCI.

The major finding from experiment 1 was that LV contractility was immediately impaired within the first hour post-SCI. Specifically, we observed that LV load-independent systolic function assessed as $E_{es}$ (Fig. 2a, b), preload-recruitable stroke work (Fig. 2d) and maximal rates of pressure generation for a given filling volume (Fig. 2e) were all reduced by 1 h post-SCI and remained depressed until 4 h post-SCI (Supplementary Table 2). We additionally examined LV contractile reserve utilizing a high-dose DOB challenge (i.e. 10 µg kg$^{-1}$ min$^{-1}$ DOB) before and 4 h following T2 SCI, and found that contractile reserve was compromised post-SCI compared to baseline (Fig. 2c). In contrast to the clear impairments to LV systolic function, LV load-independent diastolic function, as assessed with the end-diastolic pressure-volume relationship (EDPVR), was not altered acutely post-SCI (Fig. 2f). Measures of load-dependent diastolic function, including LV end-diastolic pressure ($P_{ed}$) and the rates of diastolic pressure decay (-dp/dt$_{min}$ and τ), were also unaltered from baseline in the 4 h following injury (Supplementary Table 1).

**High-dose DOB optimizes LV function and hemodynamics.** In experiment 2, we utilized a randomized and counterbalanced design whereby 14 additional animals ($n = 7$ per group) received individualized hemodynamic management with either DOB or NE starting 30 min post-SCI (Fig. 3a). Drug levels were continually titrated to achieve a target $E_{es}$ of ~2.5–2.9 mmHg ml$^{-1}$ for animals receiving DOB, based on the baseline pre-SCI mean $E_{es}$ from animals in experiment 1, and a target MAP of ~85–90 mmHg for animals receiving NE, in line with the current clinical guidelines[2]. As a result of this individualized approach, four of the animals receiving DOB were administered higher doses (i.e., ≥2.5 µg kg$^{-1}$ min$^{-1}$, DOB+) while three received negligible doses (i.e., ≤0.5 µg kg$^{-1}$ min$^{-1}$, DOB−; Fig. 3a). As such, we subsequently stratified the DOB animals by dose (i.e. DOB+ and DOB− groups). All NE animals received sufficient doses to maintain MAP at 85–90 mmHg (mean 0.16 µg kg$^{-1}$ min$^{-1}$, range 0.06–0.46 µg kg$^{-1}$ min$^{-1}$), which were similar to those reported in our group's previous studies using a low-thoracic SCI porcine model[10].

Hemodynamic management with DOB+ and NE both augmented MAP and LV contractility ($E_{es}$) up to 4 h post-SCI (Fig. 3b, c and Supplementary Tables 3 and 5), however the two drugs achieved increases to MAP via markedly different alterations to cardiac and vascular hemodynamics. Specifically, DOB+ increased MAP via improvements to LV systolic function (Fig. 3d, h) and augmented cardiac output (Fig. 3e); in contrast, NE

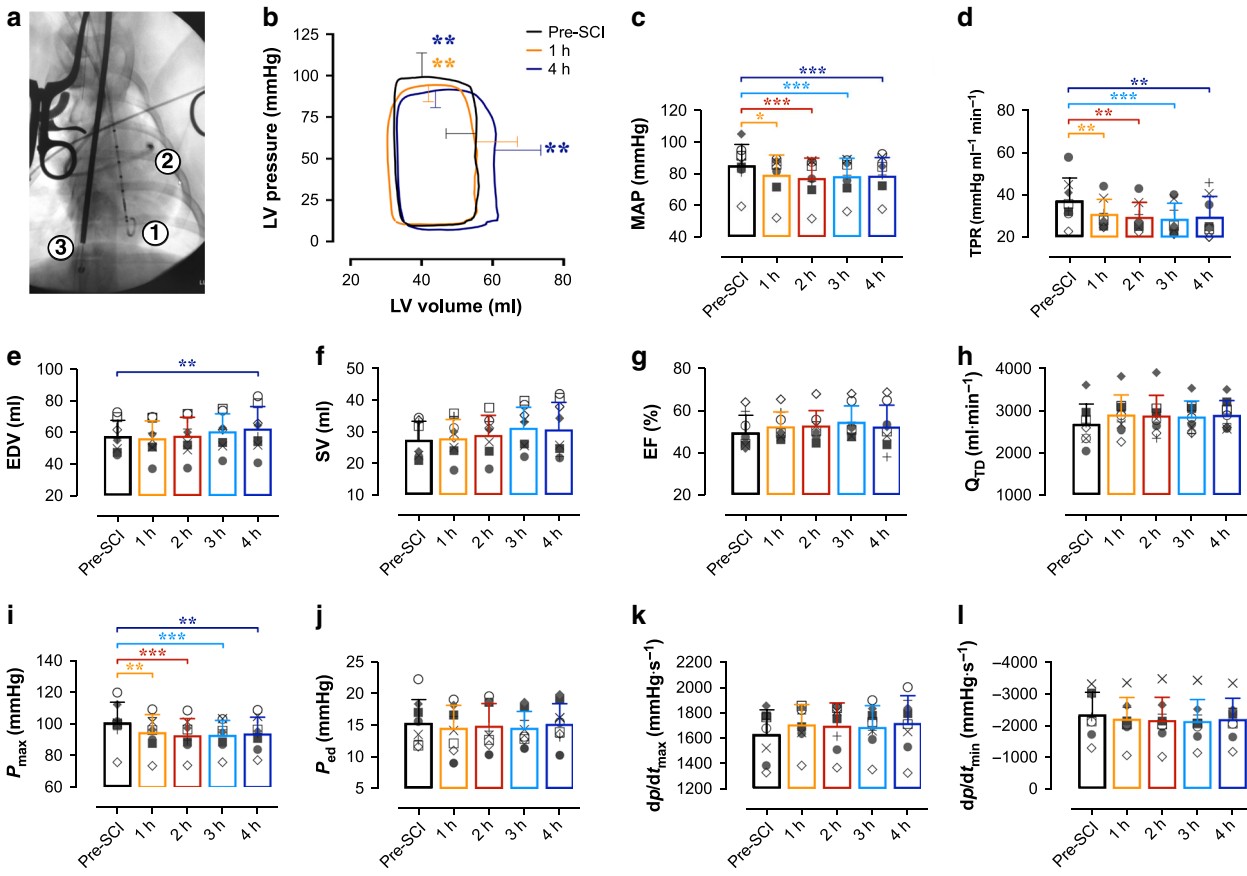

**Fig. 1 Load-dependent LV function and hemodynamics in acute T2 SCI. a** In experiment 1, $n = 8$ animals were instrumented with a LV pressure-volume admittance catheter [1], Swan-Ganz catheter [2] advanced to the pulmonary artery, and a balloon-tip inferior vena cava (IVC) occlusion catheter [3] for transient reductions to preload and assessments of LV end-systolic elastance ($E_{es}$). **b** Basal pressure-volume loops represent mean interpolated data and error bars represent standard deviation (s.d.) for peak systolic pressure ($P_{max}$) and end-diastolic volume (EDV) at baseline (pre-SCI, black), 1 h (orange) and 4 h (blue) post-SCI. $P_{max}$ was reduced within the first hour post-SCI ($p = 0.0064$) and remained lowered up to 4 h post-SCI ($p = 0.0021$). EDV was increased compared to pre-SCI at 4 h ($p = 0.0038$), though LV stroke volume (SV) was not significantly altered by the experiment endpoint. **c** Bar plots represent means, error bars show s.d., and symbols represent individual animal data. Mean arterial pressure (MAP) and **d** total peripheral resistance (TPR) were reduced within the first hour post-SCI ($p = 0.0020$ and $0.0067$, respectively), and those reductions were sustained up to 4 h post-SCI ($p \leq 0.001$ for both). **e** While EDV was augmented at 4 h post-SCI ($p = 0.0038$), **f** SV, **g** ejection fraction (EF), and **h** cardiac output ($Q_{TD}$) were unchanged post-SCI. **i** $P_{max}$ was impaired within 1 h post-SCI, but **j** end-diastolic pressure ($P_{ed}$), **k** the maximal rates of systolic pressure generation ($dp/dt_{max}$), and **l** diastolic pressure decay ($dp/dt_{min}$) were unchanged within 4 h post-SCI. *$p < 0.05$, **$p < 0.01$, ***$p < 0.001$ versus pre-SCI. Data were assessed using a one-way repeated-measures ANOVA with Fisher's LSD for post hoc comparisons versus pre-SCI data. See Supplementary Table 1 for detailed statistics. Source data are provided as a Source Data file.

augmented MAP via vasoconstrictor effects and simultaneously produced significant increases to LV afterload ($E_a$, Fig. 3g) that ultimately restricted stroke volume and cardiac output (Fig. 3d, e). DOB+ additionally enhanced LV early diastolic relaxation ($\tau$, Supplementary Table 4), which was not observed with NE despite both groups having similar heart rates throughout the experiments (Fig. 3f). DOB− animals demonstrated small but significant improvements in LV contractility ($E_{es}$), stoke work, stroke volume, and MAP; however, due to the very small doses of DOB received by DOB− animals those hemodynamic effects were minimal in comparison to DOB+ (Supplementary Tables 3–5). Though heart rate tended to increase with NE or DOB treatment (Fig. 3f), there were no significant alterations to heart rate from the treatment onset to 4 h post-SCI.

**High-dose DOB improves SCO₂ and mitigates hemorrhaging.** Within the spinal cord parenchyma, DOB+ animals exhibited large improvements to $SCO_2$ measured at the 1.2 cm probe following decompression (Fig. 4b, c), and the relative increase to $SCO_2$ was greatest in DOB+ compared to both CON and NE

animals at 3 h ($p = 0.05$ vs. NE; $p = 0.02$ vs. CON) and 4 h post-SCI ($p = 0.02$ vs. NE and CON). During the compression period (i.e., initial 2 h post-SCI), only DOB+ appeared to improve SCBF ($p = 0.028$ vs. CON at 2 h post-SCI; Fig. 4d), while SCBF remained depressed in all other animals (Supplementary Table 6). DOB+ additionally mitigated increases in the lactate-to-pyruvate ratio during the compression period (Fig. 4f and Supplementary Table 7) that were otherwise observed in the NE and CON animals.

Histological analyses at the injury epicenter demonstrated that NE exacerbated spinal cord hemorrhaging compared to CON (Fig. 4i, j), whereas animals receiving DOB+ were spared the significant increased in hemorrhaging. Immunohistochemical analyses of the injury epicentre did not reveal any between-group differences in the densities of glial fibrillary acidic protein (GFAP+) or inflammatory activation (IBA1+, Fig. 5).

**DOB treatment preserves cardiac function in chronic T2 SCI.** In experiment 3, an additional cohort of $n = 10$ animals received a T2 SCI identical to those in experiments 1 and 2, and were

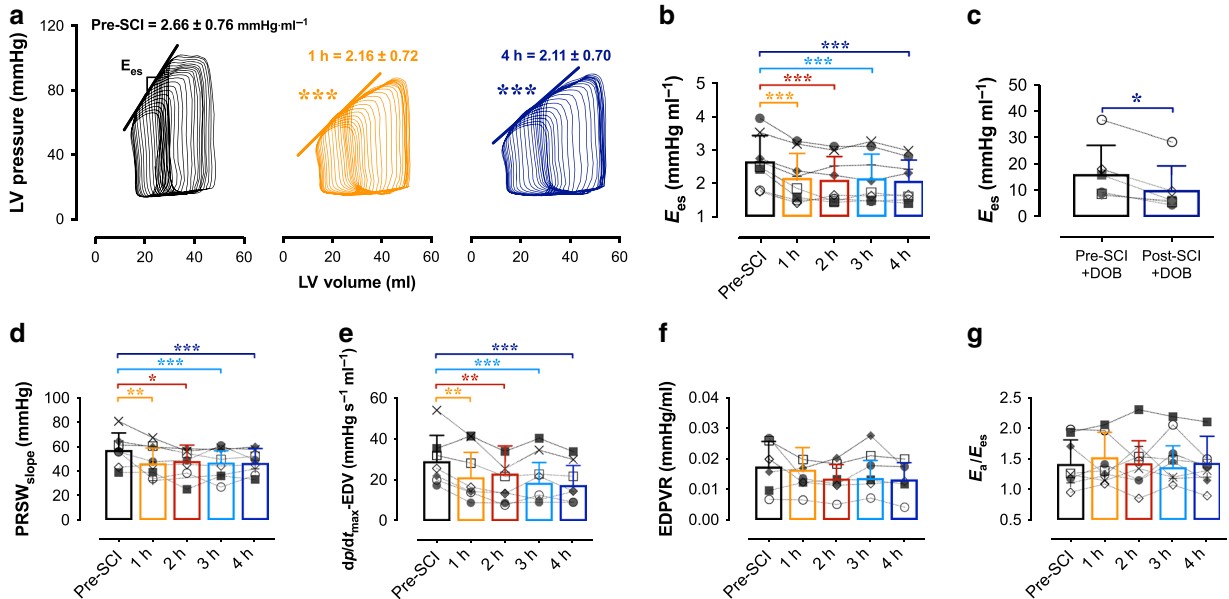

**Fig. 2 Impaired LV systolic load-independent function in acute T2 SCI. a** Representative PV loops during IVC occlusions illustrate impaired LV contractility (end-systolic elastance; $E_{es}$) within 1 h (middle, orange; $p < 0.001$) and up to 4 h (right, blue; $p < 0.001$) post-SCI. **b** Bar plots and error bars represent means and s.d., and symbols represent individual animal data. Group data show the reduction to $E_{es}$ acutely post-SCI. **c** Animals additionally had reduced $E_{es}$ responses to dobutamine challenges ('DOB', 10 µg kg$^{-1}$ min$^{-1}$) post-SCI ($p = 0.025$). **d** Impaired systolic load-independent function is further supported by the early (1 h; $p = 0.0047$) and sustained reductions to preload-recruitable stroke work (PRSW; $p < 0.001$ at 4 h vs pre-SCI) and **e** the rate of maximal pressure generation for a given EDV (dp/dt$_{max}$−EDV; $p < 0.001$ at 4 h). **f** The end-diastolic pressure–volume relationship (EDPVR) was not altered acutely post-SCI. **g** There were no changes to the relationship of arterial elastance ($E_a$) to $E_{es}$, due to simultaneous reductions in LV afterload and contractility following SCI. *$p < 0.05$, **$p < 0.01$, ***$p < 0.001$ versus pre-SCI. See Fig. 1 for statistical analyses, and Supplementary Table 2 for detailed statistics. Source data are provided as a Source Data file.

recovered and survived for 12 weeks post-SCI. On the day of SCI, animals were randomized and counterbalanced to receive continuous infusions of DOB (2.5 µg kg$^{-1}$ min$^{-1}$), NE (4.25 µg min$^{-1}$) or no treatment (CON) starting at 30 mins post-SCI until 6 h post-injury (Fig. 6a). These doses reflected the average DOB+ and NE doses delivered in experiment 2. For this experiment, a Swan-Ganz catheter was placed percutaneously into the pulmonary artery via a jugular vein, and importantly allowed us to perform repeated within-animal thermodilution measures of $Q_{TD}$ at pre-SCI, acutely post-SCI (i.e. up to 6 h) and during outcome experiments at 12 weeks post-SCI. At the experimental endpoint, cardiovascular instrumentation was identical to experiment 1 for assessments of LV load-dependent and load-independent function, and global hemodynamics.

One of the NE-treated animals suffered pulmonary complications and was euthanized 2 days post-SCI. At 12 weeks post-SCI, $Q_{TD}$ was preserved in DOB-treated animals but otherwise appeared to be lowered in both CON and NE animals (Fig. 6b). Measures during IVC occlusions indicated that LV load-independent contractility ($E_{es}$) was lower in both CON and NE groups as compared to DOB-treated animals (Fig. 6c). Animals that received NE, however, exhibited notable hypertension as indicated by uncharacteristically high MAP (Fig. 6d), total peripheral resistance (TPR, Fig. 6e) and arterial elastance ($E_a$, Fig. 6f).

## Discussion

Our findings provide compelling evidence that LV load-independent contractility is immediately impaired in acute high-level SCI, and that correcting LV contractility with DOB+ is beneficial to the spinal cord parenchyma by optimizing cord oxygenation and blood flow. We also report that acute

DOB treatment prevents the decline in cardiac function in the chronic setting post-SCI, which may have important implications for cardiovascular disease risk stratification. Furthermore, our data highlight that the current clinical standard of hemodynamic management with NE does not support improved LV function, does not modify SCBF, and appears to worsen spinal cord hemorrhaging. This research therefore supports the efficacy of implementing a cardiac-focused hemodynamic management strategy in the acute phase following high-thoracic SCI.

In experiment 1, the immediate reductions to key load-independent measures of LV systolic function incontrovertibly indicates that intrinsic contractile dysfunction in high-level SCI results from the immediate loss of descending sympathetic input to the heart post-injury. Previously, only a small collection of echocardiographic studies in humans had provided some evidence for chronically-altered LV systolic function in humans[11], and the interpretation of those findings were limited due to the load-dependent nature of echo-derived data. Our group has utilized LV pressure-volume catheterization to assess load-independent LV function in a chronic rodent model of SCI, and reported reductions to LV $E_{es}$ following a T2 or T3 injury[12,13]. Our present data extend those observations from the chronic setting by demonstrating that LV contractility is impaired immediately following high-level SCI. Importantly, we also highlight that EF was unchanged despite clear reductions to LV contractility, reinforcing that EF does not adequately detect systolic or contractile dysfunction in SCI[11]. We have further identified a reduction to LV contractile reserve acutely after the injury, which may be attributable to a rapid loss of contractility and tonic neuro-hormonal activation of the myocardium following high-level SCI. Though our group has previously reported that systolic reserve is relatively intact in chronically-injured rats with T2

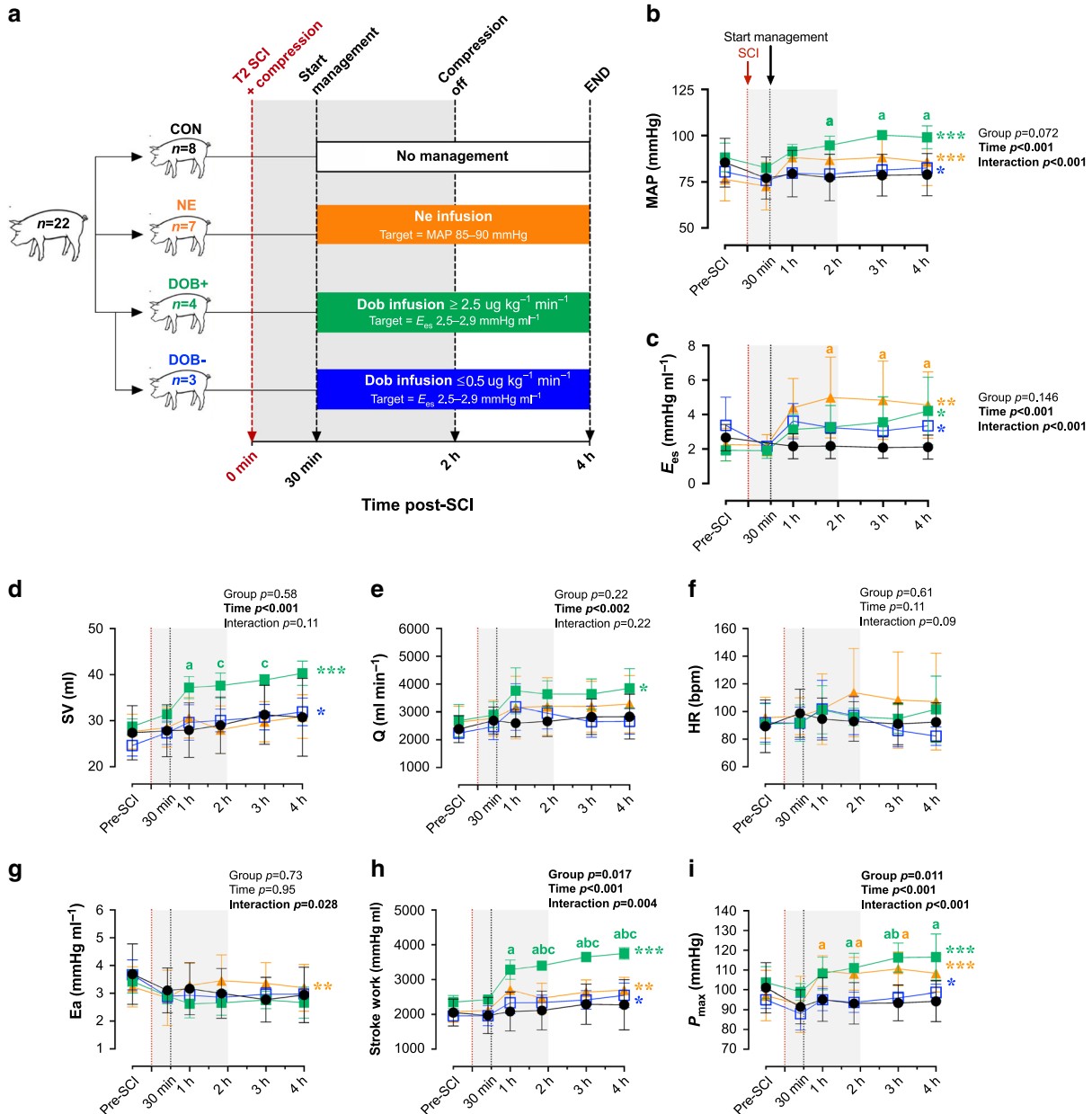

**Fig. 3 Impacts of dobutamine (DOB) and norepinephrine (NE) on LV function and hemodynamics in acute T2 SCI.** Dependent variables from 30 mins (i.e. start of hemodynamic management) to 4 h post-SCI were analyzed using a two-way repeated-measures ANOVA (factors: time, group), and post hoc comparisons were made with Tukey's test (between-group) and Fisher's LSD (within-group compared to 30 mins post-SCI). **a** Following experiment 1 (control [CON], closed circles; $n = 8$), an additional $n = 14$ animals received hemodynamic management starting 30 mins post-SCI with either DOB (squares, $n = 7$) titrated to a target $E_{es}$ of ~2.5–2.9 mmHg ml$^{-1}$, or NE (triangles; $n = 7$) titrated to a mean arterial pressure (MAP) of 85–90 mmHg. Three animals receiving DOB had a relatively high baseline $E_{es}$, and as a result received minimal doses of DOB (i.e., ≤0.5 µg kg$^{-1}$ min$^{-1}$, DOB-, open squares; $n = 3$) while $n = 4$ animals received substantial DOB doses (i.e., ≥2.5 µg kg$^{-1}$ min$^{-1}$, DOB+, closed squares). **b** Data points are means and error bars represent s.d. Both DOB+ and NE augmented MAP, however MAP was significantly elevated in DOB+ compared to CON animals. **c** LV contractility, $E_{es}$, was augmented by both DOB and NE. **d** Only DOB+ generated increases in LV stroke volume (SV) and **e** cardiac output (Q), which were not observed with NE. **f** There were no significant changes to heart rate (HR) with hemodynamic management, nor were there differences between the groups. **g** LV afterload increased with NE management but not with DOB. **h** Stroke work, an index of systolic function, is only increased with DOB+, **i** despite increases to $P_{max}$ in both NE and DOB. *$p < 0.05$, **$p < 0.01$, ***$p < 0.001$ (within-group effect for time). $^{a}p < 0.05$ vs CON; $^{b}p < 0.05$ vs DOB-; $^{c}p < 0.05$ vs NE. Within-group comparisons between 30 mins and 1 h to 3 h post-SCI are available in Supplementary Tables 3–5. Source data are provided as a Source Data file.

SCI[13],[14], those observations may result from chronic hyper-responsiveness of cardiac β-adrenergic receptors[15],[16], which would not have occurred in the acute setting of the current study. Finally, we found that heart rate was essentially unaltered up to 4 h post-SCI, mirroring data recently reported by our group in

Yorkshire pigs[9]. Although a loss of descending sympathetic input and adrenergic stimulation could be expected to result in lowered heart rates post-SCI, this is likely countered by withdrawal of vagal stimulation as cardiac vagal control remains intact after SCI[17].

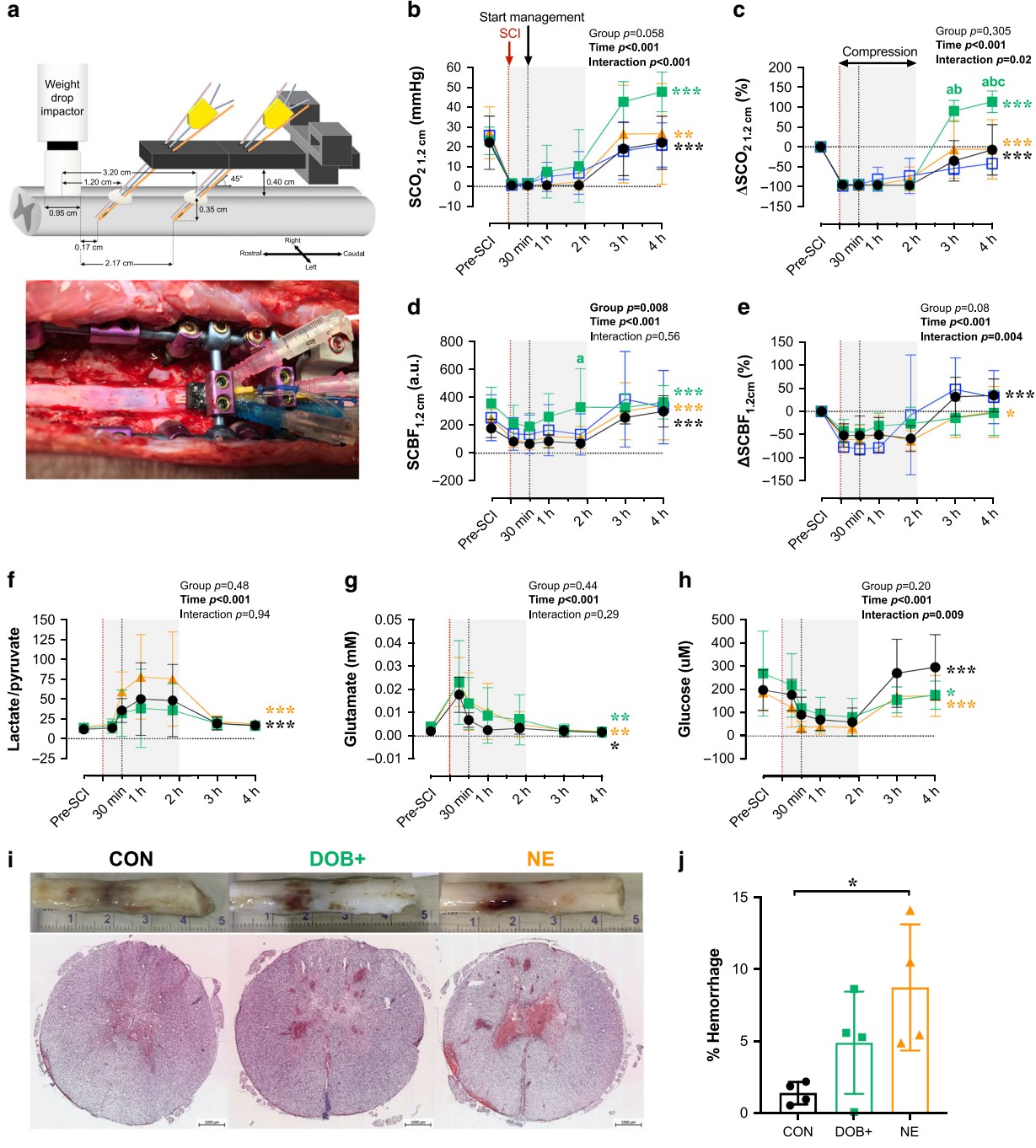

**Fig. 4 Impacts of dobutamine (DOB) and norepinephrine (NE) on the spinal cord acutely post-SCI. a** Setup for intraparenchymal monitoring. A fixation device is secured to the spinal column and probes are inserted through the dura 1.2 cm and 3.2 cm caudal to the center of the impactor. All data are shown for 1.2 cm probes. Data from 3.2 cm probes are provided in Supplementary Table 6. **b** Data points are means and error bars represent s.d. for animals receiving no treatment (control, CON; $n = 8$), high-dose DOB (DOB+; $n = 4$), low-dose DOB (DOB−; $n = 3$), and NE ($n = 7$). Improvements to spinal cord oxygenation ($SCO_2$) occur after decompression (i.e. 2 h post-SCI), but are most pronounced in DOB+. **c** When expressed as a percent change form baseline, $\Delta SCO_2$ is significantly augmented in DOB+ compared to all groups by 4 h post-SCI. **d** Spinal cord blood flow (SCBF) is augmented in CON, NE, and DOB+ following decompression, and DOB+ notably augmented absolute SCBF at 2 h post-SCI. **e** However, when expressed as percent change from baseline, SCBF was only altered in CON and NE over the treatment period. **f** The lactate/pyruvate ratio does not significantly increase after management onset (i.e. 30 mins post-SCI) with DOB+, but is increased in NE and CON. **g** In all groups, glutamate becomes progressively reduced and **h** glucose is increased following decompression (i.e. 2 h post-SCI). Sufficient microdialysis data were only acquired in $n = 2$ for DOB−, thus DOB− data were excluded from analyses. **i** Representative cords and histological stains show pronounced hemorrhaging with NE. Measures of hemorrhage were averaged over five separate sections per animal. **j** Animals receiving NE have augmented hemorrhaging at the injury epicentre, which is mitigated by DOB+. See Fig. 3 for symbol definitions and statistical details. *$p < 0.05$, **$p < 0.01$, ***$p < 0.001$. [a]$p < 0.05$ vs CON; [b]$p < 0.05$ vs DOB-; [c]$p < 0.05$ vs NE. Within-group comparisons between 30 mins and 1 h to 3 h post-SCI are available in Supplementary Tables 6 and 7. Source data are provided as a Source Data file.

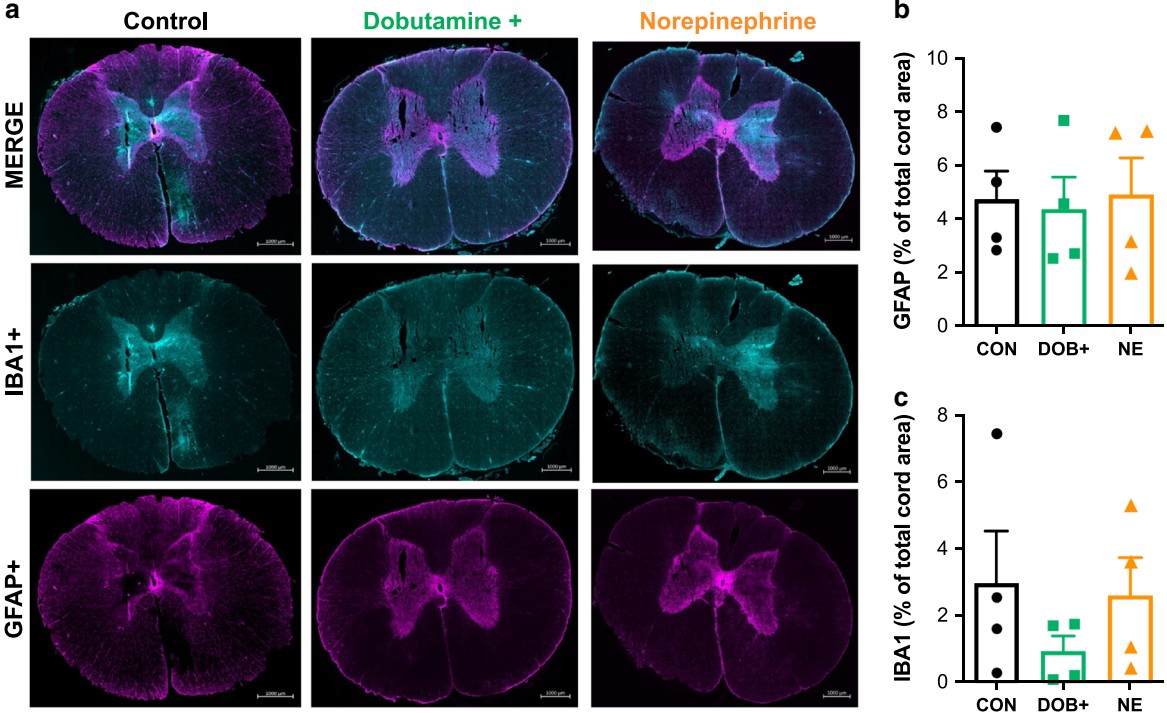

**Fig. 5 Glial and inflammatory activation in the acutely injured spinal cord epicentre. a** Representative images are shown for control (left) animals, and animals receiving hemodynamic management with high-dose dobutamine (middle) and norepinephrine (right). Merged stains (top) are shown for ionized calcium binding adaptor molecule 1 (IBA1+, middle) and glial fibrillary acidic protein (GFAP+, bottom). Group data are shown for immunohistochemical analyses of GFAP (**b**) and IBA1 (**c**). Bars plots represent means and error bars represent s.d. No significant differences were detected between animals in control (CON), high-dose dobutamine (DOB+), and norepinephrine (NE) groups. $n = 4$ per group for immunohistochemical analyses. Source data are provided as a Source Data file.

The impacts of acute high-level SCI on LV diastolic function are less clear, given there were no significant alterations to diastolic indices within the 4 h following T2 SCI. The lack of changes to the EDPVR (load-independent measure of LV compliance or stiffness) and load-dependent diastolic pressure decay is presumably a function of time, given that long-term structural remodelling and tissue stiffening generally precede diastolic dysfunction[18]. While there was a small increase to EDV by 4 h post-SCI, that may be explained by re-lengthening of the cardiomyocytes following the loss of tonic sympathetically mediated β-adrenergic activation[19]. The impacts of SCI on the diastolic phase are not well-characterized amongst the literature, though some of our group's pre-clinical work has identified blunted relaxation rates[12] alongside significant myocardial fibrosis in rodents with chronic high-level SCI[20], underscoring the time-dependency of altered diastolic function. Nonetheless, our current data suggest that diastolic function is not critically impacted within the first hours following high-level SCI.

In experiment 2, we demonstrate that correcting LV contractility with higher doses of DOB (i.e. DOB+) optimizes both cardiac and spinal cord outcomes acutely post-SCI more effectively than the current clinical standard of hemodynamic management with vasopressors (i.e. NE). Though both approaches effectively augmented MAP, this was achieved with DOB+ by increasing cardiac output, whereas NE predominantly increased vascular resistance and LV afterload ($E_a$) precluding any improvements to LV systolic output. This substantial arterial afterload is concerning as it may cause additional stress or damage to the myocardium if management is prolonged, and potentially further exacerbate the long-term negative consequences of SCI on the heart[11].

Within the spinal cord parenchyma, DOB+ appeared to alleviate spinal cord ischemia more effectively than NE by optimizing cord oxygenation, blood flow, and metabolic indices. In contrast, NE did not modify SCBF and worsened intraparenchymal hemorrhage, mirroring observations from Soubeyrand et al.[4] in a feline model of SCI. Several additional studies have linked NE with central gray matter hemorrhaging in experimental models[5,6], which is thought to result from unfavourable blood flow redistribution in the cord microvasculature[3]. Specifically, NE produces $\alpha_1$-receptor-mediated vasoconstriction that could increase resistance and reduce flow in the intact spinal cord vessels, which will divert blood flow to areas with lowest downstream pressure (i.e. the injury site) and exacerbate hemorrhage. In contrast, DOB may facilitate vasodilation of the spinal cord vasculature directly via $\beta_2$-adrenergic receptor stimulation[21] and indirectly via shear-mediated vasodilation[22], to ultimately optimize blood flow distribution and reduce hemorrhage in the injured cord. Such DOB-mediated improvements in microvascular blood flow have been reported in septic shock[23–25], and coincide with restored local oxygen delivery and mitigated tissue acidosis, all of which are in line with the $SCO_2$ and microdialysis findings in the current study. A study by De Backer et al.[24] notably highlighted that DOB-mediated improvements in capillary perfusion were not related to changes cardiac index, MAP, or total peripheral resistance, indicating that DOB may improve local microvascular flow and oxygen delivery independent of its effects on cardiac or arterial hemodynamics. In the setting of SCI, our data provide compelling evidence that DOB+ has a beneficial effect on the spinal cord parenchyma and microenvironment in comparison to NE, that may ultimately support improved microvascular perfusion and oxygen delivery to the injured cord.

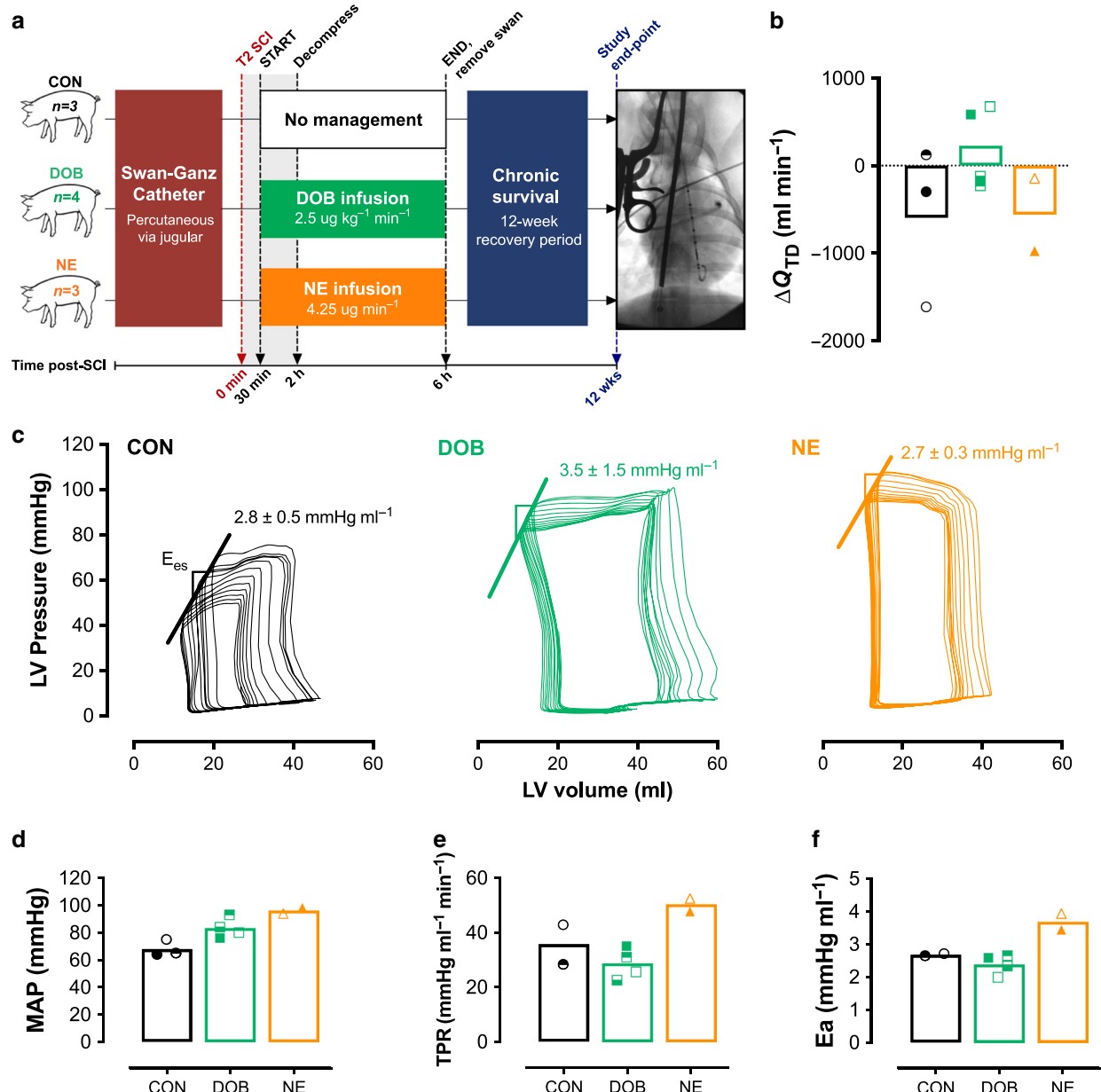

**Fig. 6 Long-term cardiovascular outcomes in animals with T2 SCI (12-week survival). a** Animals were randomized to receive no treatment (control, CON; $n = 3$), dobutamine (DOB, $n = 4$), or norepinephrine (NE, $n = 3$) treatment. Prior to injury, animals were instrumented with a Swan-Ganz catheter for thermodilution measures of cardiac output ($Q_{TD}$). Animals received a T2 SCI with hemodynamic management starting at 30 mins post-SCI and continued until 6 h post-SCI, at which point the Swan-Ganz catheter was removed. Animals were recovered and survived for 12 weeks. At study endpoint, animals were fully instrumented for cardiovascular assessments as in experiments 1 and 2. **b** All bar plots represent means, and symbols represent data from individual animals at 12 weeks post-SCI. At 12 weeks post-SCI, $Q_{TD}$ appears to be lowered in CON (circles) and animals treated with NE (triangles), but is preserved in DOB-treated animals (squares). **c** Representative left ventricular (LV) pressure–volume loops during IVC occlusions indicate superior LV end-systolic elastance ($E_{es}$) in DOB animals compared to CON and NE. **d–f** Following 12 weeks survival, mean arterial pressure (MAP) is low in animals that received no treatment (CON), but is normal in DOB-treated animals. NE animals, however, exhibit hypertension as seen with elevated MAP, total peripheral resistance (TPR), and augmented arterial elastance ($E_a$) which collectively indicate elevated cardiac afterload in those animals. Source data are provided as a Source Data file.

In experiment 3, we utilized our chronic model of T2 SCI[9] to confirm whether acute hemodynamic management with DOB can generate favourable cardiac and hemodynamic outcomes in animals survived for 12 weeks post-injury. Interestingly, we found that acute hemodynamic management with either DOB or NE appeared to prevent typical SCI-induced hypotension seen in the CON animals at 3 months post-SCI. However, only DOB-treated animals had preserved cardiac output, in addition to higher LV contractility ($E_{es}$) as compared to both NE-treated and CON

animals. Furthermore, NE animals remarkably exhibited systemic hypertension with augmented MAP (~100 mmHg) and total peripheral resistance, resulting in a near doubling of LV afterload (i.e. arterial elastance, $E_a$) versus that observed in the CON and DOB-treated animals. Such increases to afterload could lead to chronic LV pressure overload and long-term cardiac remodelling with pathological hypertrophy[26], and further limit the ability of the heart to appropriately respond to hemodynamic challenges such as exercise and/or orthostatic stress. Such changes would be

expected to confer an increased risk of cardiovascular disease, which is commonly reported in the chronic setting post-SCI[27]. Collectively, we contend that these data in a chronic model of high-level SCI complement those observed in the acute experiments, and provide further support for the efficacy of acute hemodynamic management with DOB in optimizing cardiovascular hemodynamics both acutely and chronically post-SCI.

With regards to the clinical implications of our findings, optimizing hemodynamic management continues to represent one of the only neuroprotective strategies available to clinicians to reduce secondary damage in the acute setting of SCI[1,2]. At present, there is clinical equipoise regarding what pharmacological agents provide 'optimal' hemodynamic support, and the choice of drug is most often based on the patient's hemodynamic profile[28,29], in addition to the experience the attending intensivist has with a particular approach. Our data contribute to this important area of study by providing empirical evidence that management with DOB is more efficacious than NE with respect to optimizing hemodynamics, reducing intraparenchymal spinal cord hemorrhage, and preserving cardiac function. These promising findings raise an important secondary question of whether neurological outcomes are improved with DOB treatment. Unfortunately, there are some limitations for the behavioral assessments available in the porcine model of SCI to address this question. Our group has developed a measure of hindlimb function to assess behavioral recovery in a T10 porcine model of SCI[30], but given that the injury epicentre of the T2 model is far removed from the motor pool controlling hindlimb function, it is unlikely that our approach would produce meaningful changes in hindlimb function. So, while we do not report whether DOB treatment resulted in improved hindlimb motor recovery vs. NE treatment, we contend that a focus on cardiovascular outcomes carries potentially greater clinical-relevance because patients with SCI often rank regaining autonomic function as being more important than motor recovery[28], and cardiovascular disease continues to be the leading cause of morbidity and mortality among the SCI population[27].

In conclusion, we have demonstrated that correcting the reduction in LV contractility with higher doses of DOB optimizes LV function, hemodynamics, and local cord oxygenation more effectively than the current clinical standard of NE. We therefore suggest that while DOB and NE both augment MAP, they have profoundly different effects on the injury site microenvironment, which may have important implications for long-term cardiovascular and hemodynamic function. As such, this research supports the efficacy of implementing a cardiac-focused hemodynamic management strategy in the acute phase following high-thoracic SCI.

## Methods
**Ethics and animals**. All protocols and procedures were compliant with Canadian Council on Animal Care policies, and ethical approval was obtained from the University of British Columbia Animal Care Committee (A16-0311) and United States Department of Defence (SC160098).

A total of 32 female Yucatan minipigs aged 2–3 months (20–25 kg; S&S Farms, Ramona, CA, USA) were acquired and housed in the Centre for Comparative Medicine animal facility (University of British Columbia, South Campus) for 1–2 weeks prior to surgery. Animals were housed in pairs or small groups (4–6 animals) in indoor pens with sawdust bedding and with access to an adjoining outdoor pen. Animals received daily visits from a researcher to become habituated to humans, were provided enrichment (toys e.g. chains, balls), water ad libitum and feed equal to 1.5% of body mass twice per day.

For chronic survival animals in experiment 3 ($n = 10$), all pigs received 24-h monitoring within the first 14 days following SCI, were provided continuous analgesic medication (fentanyl continuous rate infusion in the first 24–48 h, fentanyl patches thereafter), and were hand-fed and provided with water a minimum of five times per day. Animals were returned to their pens once they had sufficiently recovered from surgery, and housed in pairs in an enriched environment. Pens had sawdust bedding, padded walls, and plastic mats to prevent

slipping and scrapes. Body temperature, heart rate, bowel/bladder function, pressure ulcers, and ambulation were monitored daily over the recovery period. Animals were checked on multiple occasions throughout the day and evenings, and technicians helped to reposition the animals if they became stuck or tipped to one side. Urinary catheters were removed when animals regained the ability to void the bladder. All veterinarians and technicians were blinded to animal treatment group.

**Instrumentation and experimental timeline in experiments 1 and 2**. All animals were instrumented similarly for cardiovascular and spinal cord measurements, with bilateral pressure catheters in the femoral arteries, a PV admittance catheter placed in the LV, a Swan-Ganz thermodilution catheter placed in the pulmonary artery and an IVC occlusion catheter advanced via the right femoral vein. The PV and Swan-Ganz catheters were advanced under fluoroscopic guidance. Placement was confirmed via the emergence of a typical pressure–volume loop and typical pulmonary artery pressure waveforms. A laminectomy was then performed to expose the spinal cord from the C8-T4 level, and custom-designed sensors for spinal cord blood flow, oxygenation, pressure, and microdialysis were placed in the spinal cord parenchyma at 1.2 cm and 3.2 cm caudal to the impactor centre[31]. Once drug levels and sensors had stabilized (~2–3 h after laminectomy), baseline data for cardiac function, hemodynamics and spinal cord indices were obtained over a 30-min period. Following baseline data collection, animals received a T2 weight-drop (50 g) contusion injury with 2 h of spinal cord compression (additional 100 g, 150 g total). Cardiac, hemodynamic and spinal cord indices were continuously recorded up to 4 h post-SCI, at which point animals were euthanized and spinal cord tissue was immediately harvested.

**Effect of SCI on contractile function in experiment 1**. In $n = 8$ animals, contractile function was assessed with transient IVC occlusions for characterization of load-independent systolic function, including $E_{es}$, preload-recruitable stroke work (PRSW) and the maximal rate of pressure generation for a given end-diastolic volume ($dp/dt_{max}$-EDV), at baseline once a stable plane of anaesthesia was reached prior to SCI, and hourly post-SCI. Contractile reserve was assessed using a constant-rate infusion of DOB (10 μg kg$^{-1}$ min$^{-1}$) via an infusate port in the Swan-Ganz catheter for 10 min. This dosage has been previously utilized to challenge LV load-dependent[32] and load-independent contractile function in pigs[33]. Within the final 2 min of DOB infusion, LV $E_{es}$ was assessed with transient IVC occlusions. A minimum of 30 min recovery ($\geq 5$ half-lives of DOB[21]) was provided before 'baseline' measurements began pre-SCI, and again before euthanasia and tissue collection at the end of the experiments.

**Acute hemodynamic management in experiment 2**. For animals receiving hemodynamic management with DOB or NE ($n = 7$ per group), treatment type was randomized and counterbalanced between groups. Drugs were administered via an infusion port on the Swan-Ganz catheter, beginning at 30 min post-SCI until 4 h post-SCI. Infusions of NE were titrated to attain a target MAP of 85–90 mmHg[8], and DOB was titrated to attain an $E_{es}$ of ~2.5–2.9 mmHg ml$^{-1}$, based on the average pre-SCI $E_{es}$ observed in experiment 1. Drugs were titrated continually through the first 30 min of infusion to attain the given target, and hourly thereafter until 4 h post-SCI.

**Hemodynamic management with survival post-SCI in experiment 3**. For experiment 3 we specifically developed the methodology to place a Swan-Ganz pulmonary artery catheter percutaneously, which allowed us to repeat thermodilution measures of cardiac output on both the injury and outcome days (i.e. 12 weeks post-SCI) within the same animals. Initially, Swan-Ganz catheters were placed into a jugular vein under ultrasound guidance, and then advanced to the pulmonary artery under fluoroscopic guidance. Placement was confirmed with the pressure waveform. Laminectomy and T2 SCI were thereafter performed identically to experiments 1 and 2. A total of $n = 10$ animals were randomized and counterbalanced to receive hemodynamic management with continuous infusions of DOB (2.5 μg kg$^{-1}$ min$^{-1}$, $n = 4$), NE (4.25 μg min$^{-1}$, $n = 3$), or no treatment (CON, $n = 3$). Drug dosages were determined as the mean effective doses from DOB+ and NE animals in experiment 2. Hemodynamic management was administered from 30 min post-SCI until 6 h post-SCI, at which point the Swan-Ganz catheter was removed, and animals were recovered and housed for 12 weeks post-SCI.

At 12 weeks post-SCI, animals were anaesthetized and instrumented with a LV pressure-volume catheter, a Swan-Ganz catheter, an IVC occlusion catheter, and a femoral arterial MAP line (identical cardiovascular instrumentation to experiments 1 and 2). Once all anesthetics were maintained consistently at levels similar to the day of SCI for $\geq 1$ h, LVpressure-volume (PV) and arterial pressure data were continually recorded over a 30-min period, with serial thermodilution measures of cardiac output and IVC occlusions for assessment of LV load-independent function. Animals were euthanized when data collection was complete.

**Surgical preparation and anaesthesia**. Animals were fasted for 12 h prior to surgery, pre-anaesthetized with intramuscular injections of telazol (4–6 mg kg$^{-1}$), xylazine (1 mg kg$^{-1}$), and atropine (0.02–0.04 mg kg$^{-1}$), and thereafter induced with propofol (2 mg kg$^{-1}$). Animals received endotracheal intubation for

mechanical ventilation (10–12 breaths min$^{-1}$; tidal volume 12–15 mg kg$^{-1}$; Veterinary Anesthesia Ventilator model 2002, Hallowell EMC, Pittsfield, MA). A urinary catheter (10 F, Jorgensen Laboratories Inc., Loveland, CO) was placed for intra-operative bladder drainage, and intravenous catheters were placed for administration of anesthetic agents and fluids. A rectal temperature probe was additionally placed, and core body temperature was maintained at 38.5–39.5 °C with a heating pad (T/Pump, Gaymar Industries, Inc., Orchard Park, NY). Throughout surgery, animals received intravenous continuous rate infusions of propofol (9–13 mg kg$^{-1}$ h$^{-1}$), fentanyl (10–15 mg kg$^{-1}$ h$^{-1}$), and ketamine (5–8 mg kg$^{-1}$ h$^{-1}$), as well as intravenous fluid to maintain hydration (7–10 ml kg$^{-1}$ h$^{-1}$, 2.5% dextrose + 0.9% NaCl). The surgical plane of anesthesia was determined by the absence of jaw tone assessed by the veterinarian technicians. Standard monitoring was performed for heart rate (electrocardiogram), respiratory rate, end-tidal carbon dioxide, MAP, and oxygen saturation (pulse oximeter 8600 V, Nonin Medical Inc., Markham, ON).

**Cardiac and arterial catheterization.** After induction, animals were transferred to an operating table and oriented in a supine position. Five-centimeter incisions were made on the medial side of both hindlimbs, and tissue bluntly dissected to reveal the femoral arteries. Two-inch intravenous catheters (20 g) were advanced into the arteries and connected to fluid-filled lines. Amplifiers and pressure transducers connected the arterial lines to an A/D board (PowerLab, ADInstruments, Colorado Springs, MO) for real-time monitoring for blood pressure (i.e., systolic blood pressure [SBP], diastolic blood pressure [DBP], and MAP) with commercially available software (LabChart PRO v8.1.9, ADInstruments). Two catheters were utilized in case one of the lines failed while repositioning the animal to the prone orientation.

For placement of cardiac catheters, a 5-cm incision was made in the tissue overlaying the right jugular vein, and blunt dissection revealed the carotid artery and external jugular vein. Prior to insertion, channels for the admittance PV catheter (5F; Sciscence Catheter and ADVantage PV System [ADV500], Transonic Systems Inc.) and Swan-Ganz thermodilution catheter (7.5F; Edwards Lifesciences Canada Inc., Mississauga, ON) were connected to the A/D board for real-time visualization of catheter pressures. The pigtailed LV-PV catheter was inserted with an introducer (12 F; Fast-Cath Hemostasis Introducer, Abbott) into the carotid artery and advanced until an arterial waveform was visualized, and the Swan-Ganz catheter was inserted into the external jugular vein and advanced until a right ventricular pressure waveform became apparent. Both catheters were then further advanced under combined pressure and fluoroscopic guidance (Arcadis Avantic, Siemens Healthcare Limited, Oakville, ON) to ultimately place the PV catheter into the LV and the Swan-Ganz catheter into the pulmonary artery. Sutures were placed around the vessels and catheters to secure placement and the tissue was closed.

**Laminectomy and implantation of intraparenchymal probes.** Following placement of arterial and cardiac catheters, animals were reoriented to the prone position, and the spinous processes, laminae, and transverse processes of the C8-T7 spine were exposed with electrocautery. Using anatomical landmarks, two 3.5 × 24 mm multi-axial screws (Select™ Multi Axial Screw, Medtronic, Minneapolis, MN) were placed into the T1 and T4 pedicles. A 3.2-mm diameter titanium rod (Medtronic, Minneapolis, MN) was affixed to the screws to rigidly fix the T1-T2-T3 segments and additionally secure the weight-drop system. A T2-T3 laminectomy was performed to provide a circular window ≥1.2 cm in diameter exposing the dura mater and spinal cord, then the C8-T4 laminae were further resected to expose the spinal cord and allow for insertion of sensors and catheters surrounding the injury area.

Probes for intraparenchymal spinal cord monitoring and microdialysis were placed into their desired locations in the spinal cord using a custom-made sensor platform[10]. The platform was adjusted and secured to the spine via attachments to the titanium rods and pedicle screws. Six custom introducers were inserted through the platform at 45° angles, entering the dura at 1.2 and 3.2 cm caudal to the impactor center. Sensors for blood flow and oxygenation (combined), pressure, and microdialysis were guided through the introducers into the ventral aspect of the white matter, with final placement of the catheter tip centers ~2 mm (proximal probes) and 22 mm (distal probes) from the edge of the impactor. Placement in the spinal cord was confirmed with a commercially available ultrasound system (LOGIQ e Vet, GE Healthcare, Fairfield, CT) using a linear array 4–10 MHz transducer (8L). Cyanoacrylate glue was applied to the dural surface surrounding catheter implantation to prevent cerebrospinal fluid leakage. A minimum of 2 h was provided for intraparenchymal probe stabilization prior to the collection of baseline data.

**Spinal cord injury.** A weight-drop impactor device with an articulating arm (660, Starrett, Athol, MA) and guide rail was mounted on a metal base and secured to the T1 and T4 vertebra with the pedicle screws described above. The tip of the impactor (0.953 cm diameter), equipped with a load cell (LLB215, Futek Advanced Sensor Technology, Irvine, CA) to acquire force of impact data, was oriented vertically above the exposed dura and cord at the T2 level. The guide rail was equipped with a linear position sensor (Balluff Micropulse®, Balluff Canada Inc., Mississauga, ON) to obtain data on impactor position during the weight drop. The device was remotely operated using LabVIEW software (v16.0, National Instruments, Austin, TX), which additionally acquired real-time impactor force and

position data. Five minutes prior to injury, animals received a bolus infusion of fentanyl (7 μg kg$^{-1}$ over 1 min). The SCI was carried out by dropping a 50-g cylinder plastic weight through the guide rail from a height of ~16 cm, with another 100 g weight added immediately following the initial weight drop for a total 150 g compression. At 2 h post-SCI, the compression weight and spinal cord injury device were removed (decompression), after which pedicle screws were removed and bone wax was used to close screw holes in the vertebral body.

**Measures and analysis of cardiac and pulmonary function.** During experiments, LV pressure and volume data were continuously obtained from the LV-PV catheter, as outlined above, and all analyses of LV-PV data were performed offline using LabChart PRO software (v8.1.9, ADInstruments, Colorado Springs, CO) with the PV Loop Analysis module. Load-dependent measures of LV pressure indices (maximal pressure [$P_{max}$], minimum pressure [$P_{min}$], end-systolic pressure [$P_{es}$], end-diastolic pressure [$P_{ed}$], maximum rate of pressure generation [dp/dt$_{max}$], maximal rate of pressure decay [dp/dt$_{min}$], and time constant of diastolic pressure decay [τ]), volumetric indices (end-diastolic volume [EDV], end-systolic volume [ESV], stroke volume [SV], and ejection fraction [EF]), stroke work and arterial elastance (E$_a$) were assessed from basal PV loops over a 1-min period immediately preceding the defined measurement point (i.e. prior to IVC occlusions and thermodilution).

For the assessment of load-independent LV function, LV preload was manipulated using transient IVC occlusions at baseline (pre-SCI), 30 min post-SCI (just prior to hemodynamic management in experiment 2) and then hourly up to 4 h post-SCI in experiments 1 and 2. In experiment 1, IVC occlusions were also performed during the DOB challenge. Analysis of ~10–15 PV loops during IVC occlusions allowed for assessments of load-independent systolic and diastolic function, including $E_{es}$ and the end-diastolic pressure–volume relationship (EDPVR), respectively.

During experiments, pulmonary artery pressures were monitored in real-time and continually recorded from the Swan-Ganz catheter. Cardiac output ($Q_{TD}$) was measured with the thermodilution technique at baseline (pre-SCI), 30 min (prior to hemodynamic management), and then hourly post-SCI. Briefly, bolus infusions of iced saline (~10 ml, 0–6 °C) were administered through a temperature-recording flow-through housing (REF: 93505, Edwards Lifesciences Canada Inc., Mississauga, ON) and into the proximal port of the Swan-Ganz catheter. Bolus infusions for thermodilution were always performed ≥1 min following IVC occlusions. Calculations of $Q_{TD}$ were performed offline using LabChart PRO software (v8.1.9, ADInstruments, Colorado Springs, CO) with the Cardiac Output Analysis module (v1.3).

**Measures of intraparenchymal hemodynamics and metabolism.** Spinal cord oxygenation and blood flow were monitored in real-time using LabChart PRO software (v8.1.9, ADInstruments, Colorado Springs, CO) with a multi-parameter probe (NX-BF/OF/E, Oxford Optronix, Oxford, UK) attached to a combined OxyLab/OxyFlo channel monitor. Spinal cord partial pressure of oxygen (SCO$_2$) was measured with fiber optic oxygen sensors that utilize the fluorescence quenching technique[34]. Relative changes in SCBF were monitored with laser-Doppler flowmetry. Spinal cord pressure was assessed with custom-manufactured fiber optic pressure transducers (FOP-LS-NS-1006A, FISO Technologies Inc., Harvard Apparatus, Quebec, Canada) that employ Fabry-Pérot interferometry[10,35]. Pressure transducers were connected to a signal conditioner module (EVO-SD-5/ FPI-LS-10, FISO Technologies Inc., Harvard Apparatus, Quebec, Canada) and data were continually recorded using the Evolution software (v2.2.0.0, FISO Technologies Inc., Harvard Apparatus, Quebec, Canada).

Energy-related metabolites were measured in the spinal cord extracellular fluid with microdialysis probes (CMA11, CMA Microdialysis, Harvard Apparatus, Quebec, Canada). A subcutaneous implantable micropump (SMP-200, IPrecio, Alzet Osmotic Pumps, Durect Corporation, Cupertino, CA) was used to continuously perfuse probes with artificial cerebrospinal fluid (Perfusion Fluid CNS, CMA Microdialysis, Harvard Apparatus, Quebec, Canada) and dialysates were acquired and frozen with dry ice every 15 min, starting at baseline (pre-SCI) until 4 h post-SCI. Samples were analyzed for five metabolites (i.e., lactate, pyruvate, glucose, glutamate, and glycerol) within 1 week of collection (ISCUS$^{flex}$ Microdialysis Analyzer, M Dialysis, Stockholm, Sweden). Measures of SCO$_2$, SCBF, and spinal cord pressure as well as microdialysis samples were acquired from both locations (i.e., 1.2 and 3.2 cm caudal to the impactor) throughout experiments 1 and 2.

**Spinal cord histology and immunohistochemistry.** Following euthanasia, 6 cm of spinal tissue surrounding the injury epicenter was removed and placed in 4% paraformaldehyde. Over the next 15 days, tissue was placed in increasing concentrations of sucrose until a concentration of 30% was reached. The tissue was then cut into 1 cm sections (ensuring the injury epicenter is within a single section), then embedded in optimal cutting temperature matrix (Shandon Cryomatrix, Thermo Scientific), frozen, and kept at −80 °C. The injury section was further cut into 30 μm sections and mounted onto a series of 10 slides coated with Silane Surgipath Solution (Leica). These slides were then stored at −80 °C. For histology and immunohistochemistry, sections were thawed at room temperature for 1 h, at

which time a hydrophobic barrier was drawn using ImmEDGE Hydrophobic Barrier Pen (Vector Laboratories). Sections were rehydrated in 0.1 M PBS for 10 min then incubated with 10% normal donkey serum 0.2% Triton X-100 plus 0.1% sodium azide in PBS. Sections were incubated overnight with primary antibody rabbit anti-IBA1 (1:1000, Novus NBP2-19019). Sections were incubated for 2 h with secondary antibody Alexa Fluor 488 donkey anti-rabbit (1:1000, abcam ab150073), and glial fibrillary acidic protein (GFAP) conjugated Cy3 produced in mouse (1:1000, Sigma C9205). Slides were cover-slipped with ProLong Gold with DAPI (Invitrogen). For hematoxylin and eosin (H&E) staining, slides were thawed at room temperature and staining was conducted using standard techniques laid out by Leica. Slides were cover-slipped with ProLong Gold (Invitrogen).

Immunohistochemical staining was imaged using a Zeiss Axiophot microscope (Carl Zeiss, Oberkochen, Germany) equipped with a digital camera (Olympus Q5). H&E stains were imaged using a Leica Aperio CS2 scanner (Leica Biosystems, San Diego, CA, USA). All images were processed and analyzed by standard densitometric analyses using ImageJ (v1.52e, U. S. National Institutes of Health, Bethesda, Maryland, USA). Briefly, quantification of the immunostaining, GFAP and IBA1, was carried out by measuring the immunopositive areas in the spinal cord section. Similarly, quantification of the H&E staining was carried out by manually outlining the area of the regions of hemorrhaging, which were identified by areas that exhibited dense red staining. All positive stains are expressed as a percentage of total spinal cord area. Reported values reflect means of five separate sections per animal.

**Statistical analysis and sample size calculation.** Data are presented as means ± standard deviation (SD) in figures and Supplementary tables (for non-normally distributed data, medians and interquartile ranges are provided in Supplementary Table 8). Normalcy was determined using the Shapiro-Wilk test. For experiment 1, normally distributed dependent variables were analyzed using a one-way repeated-measures analysis of variance (ANOVA). Post hoc pairwise comparisons were made with Fisher's LSD for planned within-group comparisons, and Tukey's HSD for between-group comparisons. Paired $t$-tests were used to compare data between DOB challenges pre- and post-SCI. For experiment 2, normally distributed dependent variables were analyzed with a repeated-measures ANOVA with two independent factors (group × time), and when a significant effect was detected post hoc comparisons were performed with Fisher's LSD for within-group data, and Tukey's HSD for between-group data. For non-normally distributed data, a Friedman repeated-measures ANOVA on ranks was used to detect within-group differences over time, and within-group pairwise comparisons performed with Wilcoxon matched pairs test. For between-group comparisons, a Kruskal–Wallis ANOVA was used with Mann–Whitney $U$ tests for pairwise comparisons. All statistical analyses were performed using Statistica (v13, TIBCO Software Inc., Palo Alto, CA) with α set a priori to 0.05.

Prior to this study, there were no published data on LV $E_{es}$ in a porcine model of SCI. However, in our group's rodent studies of T2 SCI, we have reported a significant mean difference in $E_{es}$ of 0.67 mmHg μl$^{-1}$ with a pooled SD of 0.17 mmHg μl$^{-1}$ between animals with T2 SCI and sham injury (i.e., control)[13]. For experiment 1, assuming a power of 0.95 and α of 0.05 we required a minimum of six animals per group to detect significant changes in $E_{es}$ across four time-points (pre-SCI and every hour up to 4 h post-SCI). We chose to include a minimum of seven animals per group to account for any discrepancies in placing the LV-PV catheter and spinal cord probes.

For experiment 2 examining the impacts of hemodynamic management, no published data in the porcine model of SCI had reported significant between-group differences in SCO$_2$. However, our group had reported a pooled SD for SCO$_2$ (expressed as percentage of baseline) of 40% in animals receiving vasopressor-based hemodynamic management[10]. With seven animals per group, an SD of 40% and an α of 0.05, we had 95% power to detect a difference of 29% in SCO$_2$ between groups.

**Reporting summary.** Further information on research design is available in the Nature Research Reporting Summary linked to this article.

## Data availability
The data supporting the findings of this study are available from the corresponding author upon reasonable request. A Reporting Summary for this Article is available as a Supplementary Information file. Source data are provided with this paper.

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

## Acknowledgements

We thank Dr. Robert Boushel for providing his equipment and expertise for the ther-modilution method in these experiments. We are grateful for the exceptional staff of veterinarians and technicians at the Center for Comparative Medicine whose dedication and commitment to the welfare of the animals makes these technically challenging experiments possible. This work was supported by the Office of the Assistant Secretary of Defense for Health Affairs,through the Spinal Cord Injury Research Program under Award No. W81XWH 17-1-0660. Opinions,interpretations, conclusions and recom-mendations are those of the author and are not necessarily endorsed by the Department of Defense. This research was additionally funded by the Craig Nielsen Foundation (459120) and Michael Smith Foundation for Health Research (Trainee Award #17197). Dr. Kwon is the Canada Research Chair in Spinal Cord Injury and Dvorak Chair in Spine Trauma. The laboratory of Dr. West is supported by an infrastructure grant from the Candian Foundation for Innovation and the BC Knowledge Development Fund.

## Author contributions

C.R.W. and B.K.K. contributed to the conception of the study and its design, inter-pretation of data, drafting, and final editing of the manuscript. A.M.W. contributed to the study design, data acquisition, analyses and interpretation, drafting, and revision of the manuscript. N.M. contributed to the study design, data acquisition, and revision of the manuscript. E.E. contributed to data acquisition and analyses, and revision of the manuscript. K.T., K.S., K.Sh., M.W., S.F., A.B., A.M., S.T., F.S., and K-T.K. contributed to the data acquisition and revision of the manuscript.

## Competing interests

The authors declare no competing interests.
