## [Peer Review File · Nature Communications]

REVIEWER COMMENTS

Reviewer #1 (Remarks to the Author):

This study described the treatment effect after systemic dobutamine delivery in the porcine acute spinal trauma model. Overall this study provides convincing evidence that this treatment improves otherwise altered cardiac contractility and the associated reduction in spinal cord parenchymal O₂ and spinal cord blood flow. In addition, the degree of intraspinal hemorrhage was also reduced in treated animals.

From a clinical perspective this short-term post-injury data are significant as they establish a potentially new platform on how to control systemic and spinal hemodynamic changes resulting from spinal cord contusion injury. The maintenance of spinal cord blood flow and O₂ saturation is certainly one of the key contributors defining the degree of neuronal and glial cell degeneration in and around the spinal trauma epicenter.

The limitation of this study as presented in that only short post-injury period was studied. Accordingly a true clinically validated treatment effect, as defined by the degree of neurological function recovery, can not be assessed. The data on the functional neurological outcome and a corresponding reduction in spinal cord degeneration would establish the rationale for using this treatment approach/protocol in a clinical setting.

Reviewer #2 (Remarks to the Author):

The authors showed that high spinal cord injury results in a significant decrease in myocardial contractility with substantial reductions in regional blood flow and oxygenation. They showed that treating dobutamine is more efficacious at increasing local oxygen availability than norepinephrine and limited the hemorrhage in the injured cord.

The methods are sound and the observations complete, even including histopathologic evaluation. The second set of experiments reproduces a clinical approach.

Hence, the data are convincing and the clinical implications sound.

General comments

1. The authors focus on 'cardio-centric effects' of dobutamine, but the drug also has important peripheral effects, susceptible to improve the microcirculation- see e.g. the old but classical paper by De Backer et al in sepsis (Crit Care Med 2006).
2. Although the myocardial depression is very well documented, one could wonder whether dobutamine could not have protective effects in the absence of the initial decrease in blood flow, by increasing DO₂ to supranormal levels and/or by improving the microcirculation. Of course this is speculative, but it is worth a comment.
3. The constant heart rate is unexpected (Figure 3), but these data are not discussed anywhere in the text. HR is expected to decrease with the decrease in adrenergic tone, and increase with

dobutamine more than with norepinephrine. The authors should discuss it.

Specific comments

1. Abstract: the authors should indicate how SCO_2 was assessed (spinal cord partial pressure of oxygen monitored with fiberoptic oxygen sensors);
2. Abstract: it should also indicate the microdialysis measurements ('metabolism' is a bit vague);
3. Line 113: this is not really true: norepinephrine increases arterial pressure by increasing vascular tone (and this results in an increase in afterload);
4. Line 156: is there a reason to be so prudent? Yes, it is contractility ('baseline' can be deleted);
5. Line 193: 'via cardiogenic improvements in MAP' is awkward: one would not select dobutamine to increase MAP – that is how one prefers norepinephrine for more consistent effects on MAP;
6. Line 382: 'microdialysis were acquired': microdialysis is only a technique;
7. Last lines 205-206: 'These findings merit clinical investigation...': I wonder what the authors have in mind. These situations are relatively rare and the tools for monitoring quite limited. No-one would propose a RCT on this.

Reviewer #1

Feedback: *“This study described the treatment effect after systemic dobutamine delivery in the porcine acute spinal trauma model. Overall this study provides convincing evidence that this treatment improves otherwise altered cardiac contractility and the associated reduction in spinal cord parenchymal O₂ and spinal cord blood flow. In addition, the degree of intraspinal hemorrhage was also reduced in treated animals. From a clinical perspective this short-term post-injury data are significant as they establish a potentially new platform on how to control systemic and spinal hemodynamic changes resulting from spinal cord contusion injury. The maintenance of spinal cord blood flow and O₂ saturation is certainly one of the key contributors defining the degree of neuronal and glial cell degeneration in and around the spinal trauma epicenter.”*

Thank you for your thoughtful and informative comment. We are delighted that you found merit in our study, and are grateful for the helpful and constructive suggestions provided.

C1. *The limitation of this study as presented in that only short post-injury period was studied. Accordingly, a true clinically validated treatment effect, as defined by the degree of neurological function recovery, cannot be assessed. The data on the functional neurological outcome and a corresponding reduction in spinal cord degeneration would establish the rationale for using this treatment approach/protocol in a clinical setting.*

R2. We agree that an *in vivo* study demonstrating the potential long-term benefits of a cardiac approach to hemodynamic management is an important follow-up to the acute data in our manuscript. Because we were also interested in this question, we began such a study 18 months ago to test whether acute hemodynamic management with dobutamine (DOB) produces more favourable long-term cardiovascular outcomes compared to either norepinephrine (NE) or no hemodynamic management (CON). To date, we have completed data collection on 10 pigs with a T2 contusion SCI that we have survived for a 3-month period. We randomized the animals to receive DOB ($n=4$), NE ($n=3$) or no treatment (CON, $n=3$). Unfortunately, one of the animals treated with NE suffered pulmonary complications and had to be euthanized two days after injury. For this study, we developed the methodology to place the Swan-Ganz catheter percutaneously such that we could collect repeated-measures data for cardiac output in our animals on the day of injury and outcome day 12 weeks later. At the study end-point we additionally performed the same cardiac catheterization preparation as our acute study (i.e. with LV-PV, Swan-Ganz, IVC occlusion and arterial catheters). We have added these data to a new figure in the manuscript (Figure 6), and included the methodology, results and discussion of this study (Experiment 3) appropriately throughout the revised version of the manuscript.

In summary, the data from these 10 animals indicate that hemodynamic management with either DOB or NE may prevent the SCI-induced hypotension at 3 months post-injury. However, only dobutamine appears to preserve cardiac function chronically after the high-level injury. Of note, cardiac output and LV contractility were 20-30% lower in NE and CON as compared to DOB-treated animals 3 months post-SCI. Additionally, animals treated with NE were in fact relatively hypertensive (MAP=100 mmHg) with elevated vascular resistance and cardiac afterload, which may ultimately limit cardiac function and impair peripheral and central vascular function.

With regards to assessments of neurological recovery in the chronic study, we agree that this is a clinically pertinent question; however, we believe that this is not feasible in our pig model

of T2 SCI since there are no validated metrics of forelimb function in the pig. Though a measure of hind-limb function has been developed by our group for a T10 porcine model of SCI (Lee et al. 2013, J Neurotrauma), this assessment is exceptionally unlikely to be impacted by any potential neuroprotective approach in the T2 spinal injury model, since the epicentre of the injury is so far removed from the motor pool that controls hind-limb function (lumbar cord). Perhaps most importantly, the clinical decision as to how to hemodynamically manage an acutely injured patient with SCI is less based on the recovery of motor function, but rather on which approach is the most likely to favourably improve systemic hemodynamics and spinal cord oxygenation with a view to preventing chronic hemodynamic instability and tissue hypoxia. As such, we believe our focus on cardiovascular outcomes is more clinically relevant than a focus on neurological recovery in the T2 model of SCI – evidence suggests patients agree with this since they often rank the recovery of autonomic function above the recovery of motor function (Anderson 2004, Neurotrauma). To address this point, we have included a paragraph in the discussion (Lines 265-284) outlining considerations for neurological recovery, and the important clinical relevance of hemodynamic outcomes for guiding neuroprotective therapies in SCI.

Lastly, we agree that determining the degree of spinal cord degeneration will be an important focus for future work. Such an assessment will require us to develop several immunohistochemistry protocols that to our knowledge have never been performed in the pig spinal cord. We plan to actively pursue this question when we have the full complement of spinal cord tissue from all animals.

Reviewer #2

Feedback: *“The authors showed that high spinal cord injury results in a significant decrease in myocardial contractility with substantial reductions in regional blood flow and oxygenation. They showed that treating dobutamine is more efficacious at increasing local oxygen availability than norepinephrine and limited the hemorrhage in the injured cord. The methods are sound and the observations complete, even including histopathologic evaluation. The second set of experiments reproduces a clinical approach. Hence, the data are convincing and the clinical implications sound.”*

We thank the reviewer for their thorough review of the manuscript, and we are very pleased that they find our data to be convincing and clinically-relevant.

General comments

C1 + 2. *The authors focus on ‘cardio-centric effects’ of dobutamine, but the drug also has important peripheral effects, susceptible to improve the microcirculation- see e.g. the old but classical paper by De Backer et al in sepsis (Crit Care Med 2006).*

Although the myocardial depression is very well documented, one could wonder whether dobutamine could not have protective effects in the absence of the initial decrease in blood flow, by increasing DO₂ to supranormal levels and/or by improving the microcirculation. Of course this is speculative, but it is worth a comment.

R1 + 2. We absolutely agree with the reviewer that dobutamine likely improves cord oxygenation via improvements in microcirculation and/or oxygen diffusion in the primary and secondary lesion sites. We also thank the reviewer for the reference to the De Backer paper which has helped to strengthen our discussion. In the original manuscript we attempted to discuss the concept of microcirculation but agree these points require more clear articulation. As such, we have significantly revised the fourth paragraph of the discussion section (now Lines 248-264) to provide a more thorough comment on the potential beneficial impacts of Dobutamine on the cord microcirculation and oxygen diffusion.

C3. *The constant heart rate is unexpected (Figure 3), but these data are not discussed anywhere in the text. HR is expected to decrease with the decrease in adrenergic tone, and increase with dobutamine more than with norepinephrine. The authors should discuss it.*

R3. Certainly, one might expect that because HR is regulated, in part, by cardiac sympatho-adrenergic control, that HR might be reduced with a loss of descending sympathetic input to the heart following SCI. However, HR is also strongly regulated by the vagal system at rates under ~140 BPM (White & Raven 2014, J Physiol), and the cardio-vagal baroreflex does remain intact following SCI (Krassioukov & Claydon 2006, Prog Brain Res). The lack of change in HR amongst control animals with SCI is therefore not entirely surprising, as the vagal reflex likely countered any reductions in HR post-SCI that would be expected to occur due to reduced sympathetic tone. We have recently reported similar observations whereby both the T2 and T10 porcine models of SCI have relatively stable HR's from pre- to 4 hours post-SCI (West, Poormasjedi-Meibod et al. 2020, J Physiol). In the current study, we did observe HR tended to increase with NE (groupXtime interaction $p=0.09$; $p=0.062$ at 2 hrs post-SCI) but these responses were not statistically significant owing to larger amounts of

between-animal variability. We have outlined and discussed these HR data in the results and discussion:

- Line 98: “however, there were no significant alterations to LV stroke volume ... or heart rate (Supplemental Table S1a) within the 4 hours following T2 SCI.”
- Line 136-138: “Though heart rate tended to increase with NE or DOB treatment (Fig. 2f), there were no significant alterations to heart rate from the treatment onset to 4 hours post-SCI.”
- Lines 202-206: “Finally, we found that heart rate was essentially unaltered up to 4 hours post-SCI, mirroring data recently reported by our group in Yorkshire pigs {West 2020}. While a loss of descending sympathetic input and adrenergic stimulation could be expected to result in lowered heart rates post-SCI, this is likely countered by withdrawal of vagal stimulation as cardiac vagal control remains intact after SCI {Krassioukov 2006}.”

Specific comments

C1. *Abstract: the authors should indicate how SCO₂ was assessed (spinal cord partial pressure of oxygen monitored with fiberoptic oxygen sensors).*

R1. We have added a sentence to the abstract that reads “Using a porcine model of T2 SCI, we assessed end-systolic elastance as a marker of cardiac contractility via invasive left ventricular pressure-volume catheterization, and monitored intraparenchymal SCO₂ and SCBF with fiberoptic oxygen sensors and laser-Doppler flowmetry, respectively. We also quantified spinal cord metabolic markers with microdialysis”

C2. *It should also indicate the microdialysis measurements (‘metabolism’ is a bit vague).*

R2. We have included this as noted in the response to comment 1 above.

C3. *Line 113: this is not really true: norepinephrine increases arterial pressure by increasing vascular tone (and this results in an increase in afterload).*

R3. We appreciate that the wording of this statement may have been misleading. We have adjusted the wording such that lines 128-130 now read: “Specifically, DOB+ increased MAP via improvements to LV systolic function (Fig. 3d and Fig. 3h) and augmented cardiac output (Fig. 3e); in contrast, NE augmented MAP via vasoconstrictor effects and simultaneously produced significant increases to LV afterload (E_a, Fig. 3g) that ultimately restricted stroke volume and cardiac output (Fig. 3d-e)”

C4. *Line 156: is there a reason to be so prudent? Yes, it is contractility (‘baseline’ can be deleted).*

R4. “Baseline” has been deleted from this statement.

C5. *Line 193: ‘via cardiogenic improvements in MAP’ is awkward: one would not select dobutamine to increase MAP – that is how one prefers norepinephrine for more consistent effects on MAP.*

R5. We agree with the reviewer that this phrasing was awkward. We have largely revised much of the referenced paragraph in line with the reviewer's general comments above. The paragraph places a greater focus on discussing and contrasting the potential effects of DOB and NE on the spinal cord microvasculature. As such, the statement '*via cardiogenic improvements in MAP*' has been altered and the final lines of this paragraph (Lines 244-247) now read: "In the setting of spinal cord injury, our data provide compelling evidence that DOB+ has a beneficial effect on the spinal cord parenchyma and microenvironment in comparison to NE, that may ultimately support improved microvascular perfusion and oxygen delivery to the injured cord."

C6. Line 382: '*microdialysis were acquired*': *microdialysis is only a technique.*

R6. This now reads "microdialysis samples were acquired"

C7. Last lines 205-206: '*These findings merit clinical investigation...*': *I wonder what the authors have in mind. These situations are relatively rare and the tools for monitoring quite limited. No-one would propose a RCT on this.*

R7. The reviewer has made a great point, and we agree that an RCT would not be practical given that Dobutamine is readily available, widely-utilized, and approved for use in the clinical setting. We would rather like to highlight the importance consideration that clinicians might give to utilizing Dobutamine for hemodynamic management of acute SCI patients. We have revised these concluding lines (287-291), which now read: "We therefore contend that while DOB and NE both augment MAP, they have profoundly different effects on the injury site microenvironment, which may have important implications for long-term cardiovascular and hemodynamic function. As such, this research supports the efficacy of implementing a cardiac-focused hemodynamic management strategy in the acute phase following high-thoracic SCI."

REVIEWERS' COMMENTS:

Reviewer #1 (Remarks to the Author):

All my concerns were addressed and explanation on the clinical relevance of these data provided in the discussion.

Reviewer #2 (Remarks to the Author):

the paper has improved